# Allosteric Integrase Inhibitor Influences on HIV-1 Integration and Roles of LEDGF/p75 and HDGFL2 Host Factors

**DOI:** 10.3390/v14091883

**Published:** 2022-08-26

**Authors:** Parmit Kumar Singh, Wen Li, Gregory J. Bedwell, Hind J. Fadel, Eric M. Poeschla, Alan N. Engelman

**Affiliations:** 1Department of Cancer Immunology and Virology, Dana-Farber Cancer Institute, Boston, MA 02215, USA; 2Department of Medicine, Harvard Medical School, Boston, MA 02115, USA; 3Division of Infectious Diseases, Mayo Clinic, Rochester, MN 55905, USA; 4Division of Infectious Diseases, University of Colorado Anschutz Medical Campus, Aurora, CO 80045, USA

**Keywords:** HIV/AIDS, antiretroviral inhibitor, integrase, allosteric integrase inhibitor, LEDGF/p75, HDGFL2, HIV integration, nuclear speckles, speckle-associated domains

## Abstract

Allosteric integrase (IN) inhibitors (ALLINIs), which are promising preclinical compounds that engage the lens epithelium-derived growth factor (LEDGF)/p75 binding site on IN, can inhibit different aspects of human immunodeficiency virus 1 (HIV-1) replication. During the late phase of replication, ALLINIs induce aberrant IN hyper-multimerization, the consequences of which disrupt IN binding to genomic RNA and virus particle morphogenesis. During the early phase of infection, ALLINIs can suppress HIV-1 integration into host genes, which is also observed in LEDGF/p75-depelted cells. Despite this similarity, the roles of LEDGF/p75 and its paralog hepatoma-derived growth factor like 2 (HDGFL2) in ALLINI-mediated integration retargeting are untested. Herein, we mapped integration sites in cells knocked out for LEDGF/p75, HDGFL2, or both factors, which revealed that these two proteins in large part account for ALLINI-mediated integration retargeting during the early phase of infection. We also determined that ALLINI-treated viruses are defective during the subsequent round of infection for integration into genes associated with speckle-associated domains, which are naturally highly targeted for HIV-1 integration. Class II IN mutant viruses with alterations distal from the LEDGF/p75 binding site moreover shared this integration retargeting phenotype. Altogether, our findings help to inform the molecular bases and consequences of ALLINI action.

## 1. Introduction

People living with HIV (PLWH) are prescribed a cocktail of antiviral compounds, also known as combinatorial antiretroviral therapy (ART), to suppress human immunodeficiency virus 1 (HIV-1) replication. Since the mid-1990s, three drugs comprise most ART formulations, including a backbone of two nucleoside reverse transcriptase (RT) inhibitors (NRTIs) and a third compound from a separate drug class. The nature of the third compound has reflected the history of antiretroviral drug development. Protease inhibitors, which initially filled this niche, were followed by non-nucleoside RT inhibitors [1]. The advent of raltegravir, which was the first integrase (IN) strand transfer inhibitor (INSTI) approved by the US Food and Drug Administration, further impacted ART formulations [2]. Compared to raltegravir, second-generation INSTIs dolutegravir and bictegravir impart comparatively high barriers to the generation of drug resistance [3,4,5,6], and either dolutegravir or bictegravir is now generally prescribed alongside two NRTIs to treat PLWH as well as people switching from non-INSTI containing regiments [7]. Despite these advancements, resistance to second-generation INSTIs does occur [8,9,10], highlighting the need to develop new anti-IN compounds with novel mechanisms of action.

IN is the viral enzyme responsible for integrating retroviral reverse transcripts into host cell chromatin, and INSTIs block integration by binding to the IN active site in the context of the viral preintegration complex (PIC) [11,12,13,14]. Retroviral integration into host genomes occurs non-randomly, with preferences for regulatory features such as genes and/or promoter regions that are generally shared among viruses from the same *Retroviridae* genus (reviewed in [15]). HIV-1 integration in particular favors speckle-associated domains (SPADs), which are transcriptionally active, gene-dense regions [16] in proximity to nuclear speckles [17,18,19,20,21]. SPAD-proximal integration is primarily governed by two HIV-host interactions. The interaction between primate lentiviral capsid proteins and cleavage and polyadenylation specificity factor 6 (CPSF6) [19,22] enables HIV-1 PIC transport beyond the nuclear rim [23,24], and CPSF6 depletion accordingly disrupts SPAD-proximal integration [18,19,20,21]. The interaction between lentiviral IN proteins and host lens epithelium-derived growth factor (LEDGF)/p75 [25,26,27,28], which primarily directs HIV-1 to integrate into the mid-regions of active genes [29,30,31,32,33], also contributes to SPAD-proximal HIV-1 integration [18,19,20,21].

Three globular domains comprise retroviral IN proteins: the N-terminal domain (NTD), catalytic core domain (CCD), and C-terminal domain (CTD) (reviewed in [34]). The HIV-1 IN CCD is the principle LEDGF/p75 binding site [35,36,37,38], while the NTD provides additional determinants for high-affinity binding [39]. LEDGF/p75 binds IN through its IN-binding domain (IBD), which is in the C-terminal region of the protein [40,41]. LEDGF/p75 is a member of the hepatoma-derived growth factor (HDGF) protein family [42] and one other family member, HDGF like 2 (HDGFL2, also known as HDGF-related protein 2 or HRP2), contains an IBD and binds HIV-1 IN [40]. Although HDGFL2 plays little if any role in HIV-1 integration targeting in wild type (WT) cells, a subsidiary role in gene-tropic HIV-1 integration is observed in LEDGF/p75-depleted cells [43,44].

In addition to its canonical activity, HIV-1 IN harbors a second, non-enzymatic function, wherein it binds genomic RNA to regulate virus particle morphogenesis [34]. IN-RNA binding is required to incorporate the viral ribonucleoprotein complex, which is primarily composed of genomic RNA and nucleocapsid protein, into the conical capsid core. Lacking this, HIV-1 RNA and IN are prematurely degraded in infected target cells, aborting the infectious process [45,46,47]. In vitro, HIV-1 IN tetramers as compared to monomers or dimers avidly bind RNA [47], and a variety of HIV-1 IN mutant proteins have been shown to be defective for RNA binding due to defective IN tetramerization or alteration of amino acid residues that directly contact RNA [47]. The associated replication-defective HIV-1 IN mutant viruses are referred to as class II to delineate them from class I IN mutant viruses that display normal IN-RNA binding and virion particle morphology [34,48]. In addition to IN mutations, allosteric IN inhibitors (ALLINIs) disrupt IN-RNA binding [49].

ALLINIs [50], which are also referred to as LEDGINs for LEDGF site inhibitors [51], NCINIs for non-catalytic IN inhibitors [52], IN-LAIs for IN-LEDGF allosteric inhibitors [53], and MINIs for multimeric IN inhibitors [54], are a promising class of pre-clinical compounds. ALLINIs engage IN at the LEDGF/p75 binding site and accordingly suppress gene-tropic integration when cells are treated with inhibitors at the time of HIV-1 infection [54,55,56]. However, the primary mode of ALLINI action is via aberrant IN hyper-multimerization [54,57]. Drug binding to the LEDGF/p75 binding site unveils a secondary IN binding site for a CTD from a separate IN multimer [58,59], the consequences of which generate long-chain, drug-interlinked IN polymers [60]. ALLINIs more potently inhibit HIV-1 maturation as compared to integration presumably because LEDGF/p75 is better positioned to compete for drug binding to IN during the afferent as compared to the efferent phase of virus replication [61].

Our current work was inspired by several unanswered questions concerning the mechanisms and outcomes of ALLINI action. Although genic integration targeting was reduced significantly by exposing cells to ALLINIs at the time of the HIV-1 infection, integration into genes remained enriched compared to LEDGF/p75 knockout (LKO) cells [55] and to random controls [55,56]. To comprehensively address the roles of LEDGF/p75 and HDGFL2 in ALLINI-mediated integration retargeting, we have generated HDGFL2 knockout (HKO) cells and used these alongside previously described LKO cells [62] as well as cells doubly knocked out for both factors [63]. HIV-1 produced in the presence of ALLINIs has been shown to integrate into genes at normal frequencies during the subsequent round of infection [54,64]. However, localization within the nucleus can significantly impact the frequency at which genes are targeted for HIV-1 integration. For example, genes in proximity to nuclear speckles are highly preferred integration targets [18,20,21]. Herein we show that HIV-1 produced in the presence of ALLINIs is deficient for integration into SPAD-associated genes during the subsequent round of virus infection, a trait that was shared by class II IN mutant viruses. Collectively, our findings inform the mechanisms and consequences of ALLINI action, which is germane given recent descriptions of highly potent compounds [65,66,67,68,69,70], some of which have advanced to human clinical trials [66].

## 2. Materials and Methods

### 2.1. Plasmid DNAs and Reagents

Plasmids pNLX.Luc.R- [71] and pNLX.Luc.R-.ΔAvrII [72], which encode for the same single-round HIV-1_NL4-3_ variant that carries the gene for firefly luciferase in place of *nef*, were previously described. Derivative plasmids encoding W131D [38], E246A, and K236A/E246A [73] IN mutant viruses were also as described. Plasmid pCG-VSV-G, which encodes for vesicular stomatitis virus G glycoprotein (VSV-G), was also described previously [74]. ALLINI BI-D [44] was acquired from MedChemExpress (Monmouth Junction, NJ, USA).

### 2.2. Cells, Viruses, and Infections

LKO HEK293T cells were previously described [62]. Cells doubly knocked out for LEDGF/p75 and HDGFL2 (DKO, for double knockout) were generated from LKO cells via CRISPR-Cas9 using guide RNAs targeting exons 2 and 15 of the *HDGFL2* gene [63]. HKO cells were derived from WT HEK293T cells using this same CRISPR-Cas9 strategy. HEK293T cell types were cultured in a humified atmosphere in the presence of 5% CO_2_ at 37 °C in Dulbecco’s modified Eagle’s medium (DMEM) supplemented to contain 10% (*v*/*v*) fetal bovine serum (FBS), 100 IU/mL penicillin, and 100 µg/mL streptomycin. Jurkat T cells (clone E6-1), procured from American Type Culture Collection (Rockville, MD, USA), were cultured in a humified atmosphere at 37 °C in RPMI 1640 medium containing 10% FBS, 100 IU/mL penicillin, and 100 µg/mL streptomycin in the presence of 5% CO_2_.

Single-round pseudovirus carrying the gene for firefly luciferase, hereafter referred to as HIV-Luc, was prepared by plasmid DNA co-transfection as previously described [19,71,75]. In brief, WT HEK293T cells cultured in 10 cm tissue culture plates were transfected with 15 µg total plasmid DNA (pNLX.Luc.R-:pCG-VSV-G and pNLX.Luc.R-.ΔAvrII:pCG-VSV-G ratios of 9:1) using PolyJet transfection reagent as recommended by the manufacturer (SignaGen, Frederick, MD, USA). Cell supernatants at 48 h post-transfection, precleared via centrifugation, were filtered through 0.45 µm filters by gravity flow and concentrated by ultracentrifugation at 4 °C for 2 h at 26,000 rpm using a SW32-Ti rotor. Virus pellets were resuspended in DMEM, aliquoted, and stored at −80 °C. Virus yield was determined using a commercial p24 ELISA kit as recommended by the manufacturer (Advanced Bioscience Laboratories, Rockville, MD, USA). For integration site sequencing experiments, HIV-Luc was treated with Turbo DNase (Thermo Fisher Scientific, Waltham, MA, USA) at the final concentration of 0.08 U/µL prior to infection as described [19].

To determine dose response curves as a function of host factor content, BI-D or dimethyl sulfoxide (DMSO) was added to triplicate wells at the time of HEK293T cell infection essentially as described for a prior INSTI study [75]. In brief, 10^4^ cells seeded per well in 96-well plates were infected with 2 ng p24 HIV-Luc in 200 µL for 6–8 h. Virus was removed, and cells were fed fresh growth media containing the same drug concentration or DMSO as present at the start of the infection. At 48 h, cells were washed twice with phosphate-buffered saline and lysed with passive lysis buffer (Promega Corporation Madison, WI, USA) by freezing plates at −80 °C for 30 min, which was followed by heating at 37 °C for 20 min. Following centrifugation, supernatants were assessed for luciferase activity using a Berthold Technologies luminometer. Effective concentration 50%, 70%, and 95% (respective EC_50_, EC_70_, and EC_95_) values were calculated by fitting data from two independent experiments to a dose response inhibition model (four parameters) in GraphPad Prism 8 (Dotmatics Boston, MA, USA).

ALLINI-treated HIV-Luc was produced by maintaining BI-D or DMSO solvent control throughout the course of HEK293T cell transfection. Following concentration by ultracentrifugation, resuspended virus was ultrafiltered using Amicon Ultra-15 Centrifugal Filter Units with 3 kDa molecular cut-off (Millipore Sigma, Burlington, MA, USA) to remove remaining unincorporated BI-D from the media. The concentration of HIV-Luc in the retentate was determined by p24 ELISA.

For integration site sequencing, 4 × 10^5^ cells were infected with 400 ng HIV-Luc in each well of a 6-well plate. Virus-containing media was replaced with fresh media after 6–8 h, and approximately 20% of the cell culture was harvested at 2 d post-infection to determine luciferase activity. For this, cells resuspended in passive lysis buffer were frozen overnight at −80 °C, heated at 37 °C for 30 min, and then centrifuged at 17,500× *g* for 8 min. Relative light units (RLUs) of cell supernatants were determined in triplicate by luminometer. RLU results were normalized to total protein concentration in the cell extract as determined by the Pierce bicinchoninic acid (BCA) protein assay kit (Thermo Fisher Scientific). The remaining cell culture was lysed for genomic DNA preparation at 5 d from the start of infection.

### 2.3. Western Blotting

Immunoblotting was carried out essentially as previously described [19,33]. In brief, protein concentration in cell extracts was determined by the BCA protein assay kit and 20 µg total protein was fractionated through 12% polyacrylamide gels under denaturing conditions. Gels were transferred to polyvinylidene fluoride membranes at 20 V for 30 min by electrophoretic transfer cell (Bio-Rad, Hercules, CA, USA). The following antibodies were used to detect protein signal: anti-LEDGF/p75 (A300-848; Bethyl Laboratories, Montgomery, TX, USA), anti-HRP2 (A304-314A; Bethyl Laboratories), horseradish peroxidase-conjugated anti-actin (A3854-200UL; Sigma-Aldrich, St. Louis, MO, USA), and horseradish peroxidase-conjugated anti-rabbit IgG (P5100; Agilent Dako, Santa Clara, CA, USA).

### 2.4. Preparation of Integration Site Libraries

DNA libraries were prepared by ligation-mediated PCR (LM-PCR) essentially as previously described [76,77]. In brief, isolated genomic DNA (2–10 µg) was digested with MseI and BglII restriction endonucleases overnight at 37·°C. Following purification, the DNA was ligated overnight at 12 °C to asymmetric linkers containing 5′-TA overhangs for compatibility with MseI-digested ends. Following purification, the DNA was subjected to semi-nested PCR using primers that anneal to the U5 end of HIV-1 DNA and the linker. The linker-specific primer and the second round U5-specific primer were megaprimers that contained additional sequences for Illumina clustering and sequencing. Purified LM-PCR products were subjected to 150 bp paired-end Illumina sequencing at the Dana-Farber Cancer Institute Molecular Biology Core Facilities (Boston, MA, USA).

### 2.5. Integration Site Determination and Mapping

Illumina raw reads were scanned, and integration sites were determined as described previously [19,76,77,78]. In brief, U5 and linker-specific sequences were trimmed from Illumina raw read1 and read2, respectively. Trimmed reads containing host DNA were aligned to human genome build hg19 downloaded from the UCSC server (http://genome.ucsc.edu (accessed on 1 July 2018)) by BWA-MEM aligner with paired-end option [79]. Aligned reads were filtered by SAMtools [80] and converted into BED format as described previously [19,78]. Raw Illumina sequences for viruses produced in the presence of ALLINI CX014442 accessed using Sequence Read Archive (SRA) number SRP157991 [64] were downloaded to bioinformatically determine sites of HIV-1 integration.

Integration sites were analyzed by BEDtools (command intersect) [81] to assess HIV-1 provirus distribution with respect to human genome annotations such as RefSeq genes and SPADs [18,19,20,21]. Results were compared to random integration control (RIC) values, which were generated in silico in two different ways to match utilized genome shearing strategy. RIC values based on digestion with MseI and BglII enzymes were described previously [19] while RIC values for DNA sonication, which Vansant et al. [64] used for shearing, were based on previously described fragments [20] that herein were mapped to hg19 using BEDtools. SPAD-associated RefSeq genes were identified by BEDtools [81] by quantifying the overlap between gene and SPAD coordinates; non-overlapping genes were termed SPAD-non-associated genes. For each gene set, percent integration was determined by BEDtools [81].

### 2.6. Statistical Analyses

Statistical significance in virus infection assays was assessed using two-tailed equal variance *t* test in Excel. Differences in integration site usage between samples was determined using Fisher’s exact test in Python. *P* values less than 0.05 were generally considered to be statistically relevant.

## 3. Results

### 3.1. Research Strategy

ALLINIs can perturb different aspects of HIV-1 replication. When present in target cells during the early phase of infection, ALLINIs can inhibit the overall level of integration and suppress the frequency at which genes are targeted [54,55,56,82]. ALLINIs engage the HIV-1 IN CCD dimer interface at a location coincident with LEDGF/p75 and HDGFL2 binding [36,40,51,53,54,61,82,83,84,85], but the roles of these virus–host interactions in ALLINI-mediated integration retargeting have not been systematically investigated. To address this information gap, we created HKO (for HDGFL2 knockout) cells and infected these alongside isogenic WT, LKO, and DKO HEK293T cells (Figure 1A) [62,63]. When present in virus producer cells during the late phase of replication, ALLINIs inhibit particle maturation and the resulting eccentric particles are defective for reverse transcription during the subsequent round of virus infection [52,53,54,61,85,86,87]. Although such virions reportedly retain normal gene targeting frequencies [54,64], we have further investigated this aspect of ALLINI action by stratifying genes based on established targeting preferences [18,20,21].

In the following sections, integration site data is presented in graphic and table formats. Statistical outcomes of virus infection and integration sample comparisons are presented as Appendix A.

### 3.2. The Roles of LEDGF/p75 and HDGFL2 in ALLINI-Mediated HIV-1 Integration Retargeting

We have utilized BI-D, which is a prototypical quinoline ALLINI [44,61] previously shown to reduce genic HIV-1 integration targeting in HEK293T cells [55]. BI-D EC_50_ values to inhibit the early versus late phase of HIV-1 replication are ~2.4 µM [44,53] and 0.9 µM [61], respectively, and previous work showed that LEDGF/p75 depletion significantly increased BI-D’s potency to inhibit the early phase of infection [44,61]. In order to ascertain drug effects on integration site targeting across our isogenic set of HEK293T cells, we accordingly first derived BI-D dose response curves for each cell type. WT and HKO cells were infected with single-round HIV-Luc reporter virus in the presence of a BI-D concentration range that varied from ~0.16 µM to 20 µM, while LKO and DKO cells were treated with an appropriately adjusted concentration range that varied from ~0.02 µM to 5 µM. The level of infection at each drug concentration was percent normalized to parallel cell cultures that were infected in the presence of the DMSO solvent control. As expected, BI-D potency noticeably increased in cells lacking LEDGF/p75, with calculated EC_50_ values of 1.5 µM, 1.26 µM, 0.16 µM, and 0.33 µM in WT, HKO, LKO, and DKO cells, respectively (Figure 1B). From these data, cell type-matched BI-D EC_70_ and EC_95_ values were calculated.

We next determined sites of HIV-1 integration in cells infected in the presence of cell type-adjusted EC_70_ and EC_95_ levels of BI-D and compared these results to infections conducted under baseline conditions (in the presence of DMSO). To address data reproducibility, we report side-by-side results of two independent integration site sequencing experiments. In the absence of drug, HKO, LKO, and DKO cells supported about 61%, 9%, and 3% of the level of infection supported by WT HEK293T cells, respectively (Figure 2A, DMSO). Infections conducted in the presence of EC_70_ and EC_95_ BI-D concentrations expectedly reduced these baseline values by ~70% and 95%, respectively (Figure 2A; see Appendix A for comprehensive statistical comparisons).

HIV-Luc at baseline integrated into a human gene in WT and HKO cells about 82% of the time, which expectedly decreased significantly to about 65% (*p* < 10^−26^) and 55% (*p* < 10^−54^) genic integration in LKO and DKO cells, respectively (Figure 2B, Table 1 and Appendix A). In the presence of BI-D, HIV-Luc integrated into genes about 66% to 70% of the time in WT and HKO cells, which were highly significant differences from baseline (*p* < 10^−14^). While EC_70_ BI-D concentration did not noticeably affect the basal level of gene-tropic integration in LKO cells, EC_95_ BI-D reduced this significantly, to about 56% (*p* ≤ 0.006). This level of genic integration targeting was notably similar to the level observed in DKO cells under basal infection conditions (*p* ≥ 0.7). While EC_95_ BI-D further reduced genic integration by two to three percentage points in DKO cells, these differences did not attain statistical significance versus the basal DKO cell condition (*p* = 0.3). Based on these data, we conclude that inhibition of IN binding to LEDGF/p75 and HDGFL2 in large part accounts for the ability of ALLINIs such as BI-D to retarget integration away from genes when the drugs are present during the early phase of HIV-1 infection.

### 3.3. ALLINI Treated Virions Are Defective for Integration into SPAD-Associated Genes

The requirement for specific gene KOs restricted the infection experiments in the prior section to HEK293T cells. In the following sections, drug-treated viruses were used to infect WT cell types. To address the generality of the integration targeting phenotypes, T cell lines were included in these experiments. Virions produced in the presence of 600 nM BI-D, which equates to an ~EC_95_ concentration under this condition of drug exposure [61], were initially used to infect HEK293T cells and Jurkat T cells in the absence of any added BI-D in the cell culture media. BI-D-treated HIV-Luc infected HEK293T and Jurkat T cells at approximately 6.6% and 3.3%, respectively, of the level of infection observed with viruses produced in the presence of DMSO (Figure 3A).

HIV-Luc produced in the presence of BI-D integrated into genes at lower frequencies than did the DMSO control virus (Figure 3 and Table 2). Across experiments and cell types, these differences amounted to ~1.1 to 2.6%, which in all cases failed to attain statistical significance (*p* ≥ 0.06; Appendix A). These data are consistent with prior reports that HIV-1 made in the presence of ALLINI GS-B [54] and CX014442 [64] displayed baseline frequencies of genic integration targeting.

We recently determined that SPAD-associated genes are some of the most highly targeted genes for HIV-1 integration in the human genome [21]. We accordingly next stratified the genes that were targeted for integration in Figure 3 into SPAD-associated versus SPAD-non-associated, and replotted integration into these gene subsets (Figure 4). As expected, HIV-1 highly favored SPAD-associated genes for integration: although these genes comprise but 3.3% of the human genome, nominally one-third of all integration events occurred within them (Figure 4A,C, Table 2; *p* < 10^−300^, Appendix A). Integration into SPAD-non-associated genes was also enhanced compared to random, though this approximate 1.2-fold enrichment paled in comparison to the >10-fold enrichment for integration into SPAD-associated genes (Figure 4B,D, Table 2).

HIV-Luc made in the presence of BI-D displayed statistically significant ~4% to 8% reductions in integration into SPAD-associated genes in both HEK293T cells and Jurkat T cells (Figure 4A,C, Table 2; *p* ≤ 0.017, Appendix A). Reciprocally, integration into SPAD-non-associated genes witnessed ~2% to 6% upticks in HIV-1 integration site targeting, which in three of four cases were statistically significant differences (Figure 4B,D, Table 2 and Appendix A). This inverse relationship follows the observation that bulk genic integration was unaffected under these conditions (Figure 3).

To ascertain the generality of these findings, we next reanalyzed previously reported integration site data from viruses that were produced in the presence of ALLINI CX014442 [64]. In this study, SupT1 cells were infected with HIV-1 following exposure to a CX014442 concentration range that varied from 31.25 nM to 250 nM, which spanned from less than the compound’s EC_50_ to greater than its EC_90_ (respective 69 nM and 114 nM values [88]). As per the original paper [64], we report integration sites for infections initiated with 1-to-20 and 1-to-40 dilutions of virus (Table 3 and Figure 5).

CX014442-treated virus integrated into human RefSeq genes similarly as the baseline DMSO treated virus, though we do note a statistically significant ~4.5% reduction (*p* = 0.004) in genic integration for virus made in the presence of 125 nM compound that was diluted 40-fold prior to infection (Figure 5A, Table 3 and Appendix A). In a largely dose-dependent manner, by contrast, integration into SPAD-associated genes was consistently and significantly decreased by CX014442 treatment. For example, preexposure to 125 nM and 250 nM CX014442 reduced HIV-1 integration into SPAD-associated genes by ~8% to 10% (*p* ≤ 0.0001) (Figure 5B and Appendix A). Akin to the results observed for BI-D (Figure 4B,D), these changes were accompanied by meaningful upticks in integration into SPAD-non-associated genes (Figure 5C, Table 3 and Appendix A).

### 3.4. Class II HIV-1 IN Mutant Viruses Are Defective for Integration into SPAD-Associated Genes

ALLINI treated viruses and class II HIV-1 IN mutant viruses share some commonalities, including disruption of IN binding to genomic DNA and defective virus particle morphogenesis [34,47,49,52,87], the consequences of which lead to premature IN and RNA degradation and abortive reverse transcription [34,45,46,47]. We accordingly next analyzed a small number of class II IN mutant viruses to see how their integration targeting preferences compared to WT viruses made in the presence of ALLINIs.

The class I and class II monikers are generally reserved for HIV-1 IN mutant viruses that display little-to-no residual infectivity. However, characterization of reverse transcription phenotypes can help to distinguish class I versus class II characteristics of partially defective IN mutant viruses as well. For example, if normal DNA synthesis is accompanied by an increase in 2-long terminal repeat circle formation, the mutant is categorized as class I. If by contrast the infection defect is accompanied by a DNA synthesis defect, then the mutant is typed as class II [73,89,90]. Because our goal was to map integration sites, our experimental strategy required residual levels of HIV-1 infectivity. Moreover, because LEDGF/p75 binding is mediated via the IN NTD and CCD [36,39], we selected partially infectious IN CTD mutant viruses, including E246A and K236A/E246A [73]. As a control, we in parallel analyzed the class II IN CCD mutant virus W131D [38]. Trp131 maps to the LEDGF/p75 binding site [36,37,38] and the W131D substitution partially disrupted IN-LEDGF/p75 binding [38].

In replicate experiments, IN mutant viruses K236A/E246A, E246A, and W131D infected HEK293T cells at ~3.4%, 19.8%, and 1.9% of the level of WT HIV-Luc, respectively (Figure 6A). Frequencies of IN mutant K236A/E246A and E246A viral integration into RefSeq genes varied from the WT by at most 0.8%, which were across the board statistically insignificant differences (Figure 6B, Table 4 and Appendix A). The approximate 10–12% reductions in RefSeq gene integration observed for the IN W131D mutant virus by contrast were highly significant (*p* < 10^−30^).

In contrast to total RefSeq genes, the IN CTD class II mutant viruses were generally defective for integration into SPAD-associated genes. In the case of the double mutant virus, whose residual ~3.4% infectivity mirrored the infectivity of BI-D-treated virus (Figure 3A), both experimental replicates revealed highly significant reductions (*p* < 10^−5^), with reciprocal upticks in integration into SPAD-non-associated genes (Figure 7A,B, Table 4 and Appendix A; *p* < 10^−4^). The IN E246A IN mutant virus revealed this same phenotype in one of two experimental replicates. Thus, while the frequency of SPAD-associated gene integration was statistically indistinguishable from the WT in replicate 2 samples, the significant ~3% reduction for the E246A IN mutant virus in replicate 1 (*p* = 3.6 × 10^−6^) was accompanied by a significant uptick in integration into SPAD-non-associated genes (*p* = 5.2 × 10^−5^). As expected from the overall reduction in RefSeq gene targeting, the IN W131D mutant virus was also defective for integrating into SPAD-associated genes, with one of two replicates revealing a significant uptick in integration into SPAD-non-associated genes (Figure 7A,B, Table 4 and Appendix A).

## 4. Discussion

Our work informs important aspects of ALLINI functionality. First, competition for binding of LEDGF/p75 and HDGFL2 host factors to the IN CCD appears to in large part account for the ability of these compounds to suppress genic integration targeting during the early phase of HIV-1 replication (Figure 2). Due to this retargeting, proviruses formed in the presence of ALLINIs are transcriptionally compromised [56,65], leading to the suggestion that ALLINIs may help to suppress the formation of the replication-competent latent HIV reservoir [91]. Although currently uncertain how the IN inhibitors might be effectively deployed as such “block-and-lock” agents, our work nonetheless helps to understand the underlying biology behind the integration retargeting phenotype.

Proviruses formed via ALLINI-treated virions are also compromised transcriptionally [64], and our work additionally clarifies that genic integration targeting is disrupted for viruses exposed to ALLINIs during the late phase of HIV-1 replication. Although integration into bulk RefSeq genes was largely unperturbed, integration into SPAD-associated genes, which are naturally highly preferred for HIV-1 integration [21], was disrupted (Figure 3, Figure 4 and Figure 5). These downturns were moreover generally met with significant upturns in integration into the reciprocal gene set, i.e., SPAD-non-associated genes. Somewhat unexpectedly, we observed this same phenotype for the class II IN CTD K236A/E246A double mutant virus, and for one of two experimental replicates with its more infectious single missense mutant E246A variant (Figure 6 and Figure 7). Although the molecular basis of the integration retargeting phenotype shared by ALLINI-treated and class II IN mutant viruses is unclear, different possibilities can be envisioned.

HIV-1 IN in solution exhibits concentration-dependent tetramerization (reviewed in [34]) and IN tetramers as compared to lower order monomer/dimer forms avidly bind RNA in vitro [47]. Although not investigated directly for the K236A/E246A or E246A mutants, prior work revealed that K236E mutant IN protein was primarily dimeric, and that the associated mutant viral IN was defective for genomic RNA binding [47]. We accordingly suspect that both K236A/E246A and E246A mutant INs would also be defective for IN tetramerization and RNA binding in virions, though perhaps proportionally less so than the highly defective K236E IN mutant (<1% residual infectivity [73]). Although LEDGF/p75 binding stabilizes IN tetramers [39,92], there is little reason to suspect loss of LEDGF/p75 binding as a contributing factor to IN CTD mutant viral integration retargeting. First, the NTD and CCD mediate LEDGF/p75 binding, with no apparent role for the CTD [39]. Second, the integration retargeting phenotype of the W131D control virus, whose IN is nominally defective for LEDGF/p75 binding [38], was notably different from the CTD mutant viruses (Figure 6 and Figure 7).

One limitation of our study is that the most interesting phenotypes were observed under conditions of severe HIV-1 restriction imposed by comparatively high ALLINI doses (Figure 2, Figure 4 and Figure 5) or IN mutations (Figure 7). Under these conditions, just a small percentage of the virus populations, compared to controls, remains active. ALLINIs and class II IN mutations elicit strikingly similar eccentric HIV-1 particles with the viral ribonucleoprotein complex located outside of electron-lucent cores [52,53,54,61,85,86,87], and such virions are apparently replication-defective due to premature degradation of IN and HIV-1 RNA in nascently infected cells [45,46,47]. Mishappened/deformed cores with associated electron density are formed at near equal frequencies in WT, ALLINI-treated, and class II mutant virions [87]. If such viruses are infectious, their malformed capsid lattices could potentially lead to premature uncoating in the nucleus, the consequences of which could arrest nuclear penetration and lead to integration into genes more distal from nuclear speckles than would otherwise occur during baseline infection. Image-based assays that track positions of intranuclear uncoating [93,94] may help to lend evidence in support of such a model. CPSF6 has been described as a master regulator of intranuclear HIV-1 penetration [95] and, perhaps not unsurprisingly, HIV-1 integration into SPAD-associated genes is for the most part obliterated in CPSF6 knockout cells [21].

Although not statistically relevant differences, we consistently observed minor reductions in integration into RefSeq genes in DKO cells in the presence of EC_95_ concentration of BI-D (Table 1 and Figure 2). Although it may be tempting to speculate that ALLINI treatment imparts aberrant IN hyper-multimerization in the absence of LEDGF/p75 and HDGFL2 during the early phase of HIV-1 infection, binding to nucleic acid shields IN from the ALLINI-induced effect [50,82], indicating this may not be at play in DKO cells, which support normal levels of reverse transcription [44]. Moreover, LEDGF/p75 and HDGFL2 are the only members of the HDGF family that harbor an IBD [40]. Based on the ALLINI-IN contacts that drive aberrant IN hyper-multimerization [58,59,60], it seems possible that ALLINIs could also disrupt host factor interactions with the IN CTD. Numerous host factors, including the histone acetyltransferase EP300, have been shown to bind the IN CTD [96]. Additional research is required to determine if EP300 or perhaps other CTD-binding host factors play roles in HIV-1 integration targeting and, if so, whether these effects are disrupted by ALLINI treatment.

## Figures and Tables

**Figure 1 viruses-14-01883-f001:**
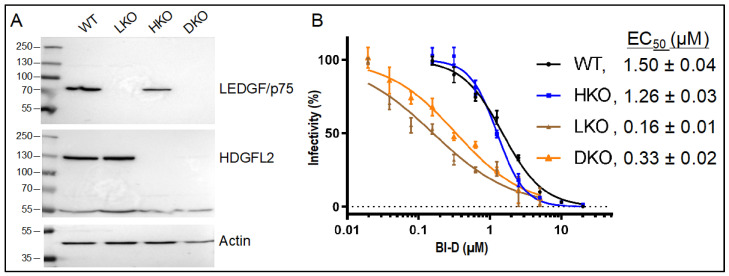
Characterization of LEDGF/p75 and HDGFL2 KO cells and BI-D dose response curves. (**A**) Western immunoblots using anti-LEDGF/p75 and anti-HDGFL2 antibodies; actin was used as a loading control. The analyzed HEK293T cell type is indicated atop the upper blot. WT, wild type; LKO, LEDGF/p75 knockout; HKO, HDGFL2 knockout; DKO, double knockout. Numbers to the left indicate positions of mass markers in kDa. Results are representative of those observed in three independent experiments. (**B**) BI-D dose response curves using the indicated cell types. Calculated EC_50_ values are shown (average ± standard deviation for *n* = 2 independent experiments; data points within each experiment were derived from technical triplicate samples).

**Figure 2 viruses-14-01883-f002:**
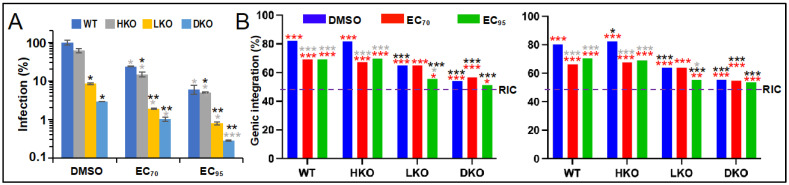
Consequences of ALLINI action in LEDGF/p75 and HDGFL2 knockout cells. (**A**) HIV-Luc infectivities in the indicated cell type in the presence or absence of BI-D. Results (average ± standard deviation from two independent experiments; within each experiment, luciferase values were determined for technical triplicate samples) were percent normalized to DMSO-treated WT cells. (**B**) Percent genic integration targeting for the infections shown in A; see Table 1 for plotted values (the graph on the left is replicate 1 data) and Appendix A for pairwise statistical comparisons. *, *p* < 0.05; **, *p* < 0.001; ***, *p* < 0.0001 (gray asterisks, versus matched DMSO control; black asterisks, versus matched WT cell condition; red asterisks, versus random integration control [RIC; dotted line]).

**Figure 3 viruses-14-01883-f003:**
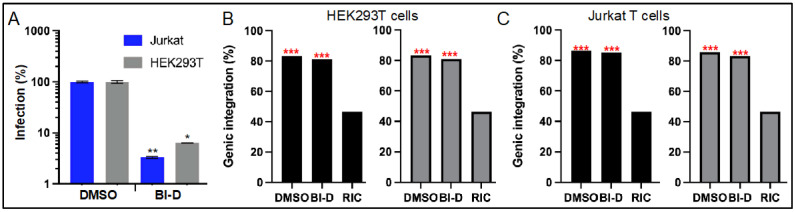
Genic integration profiles of HIV-Luc produced in the presence of BI-D. (**A**) HIV-Luc infectivities in HEK293T cells and Jurkat T cells following exposure to 600 nM BI-D or DMSO during virus production. Results (average ± standard deviation from two independent experiments; within each experiment, luciferase values were determined for technical triplicate samples) are percent normalized to the indicated DMSO-treated cell type. *, *p* < 0.01; **, *p* < 0.001 (Appendix A). (**B**) Percent integration in RefSeq genes in HEK293T cells for the infections shown in panel (**A**). The left graph shows results for replicate 1. (**C**) Same as in panel (**B**), except for Jurkat T cells—see Table 2 for plotted values and Appendix A for pairwise statistical comparisons. ***, *p* < 0.0001 versus random integration control (RIC).

**Figure 4 viruses-14-01883-f004:**
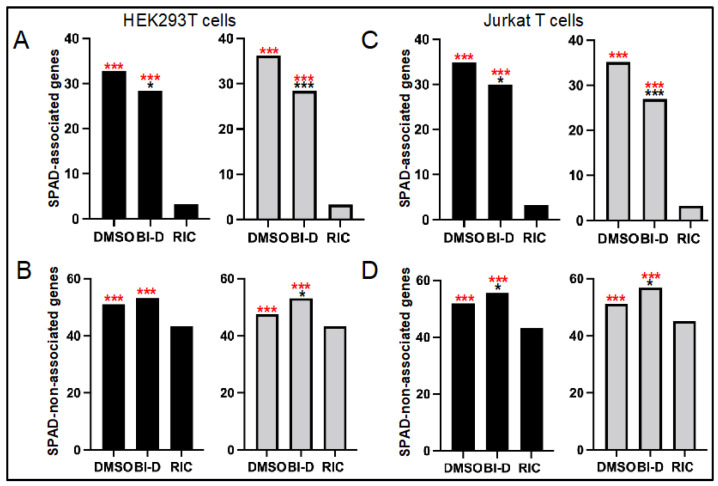
HIV-Luc produced in the presence of BI-D is defective for integration into SPAD-associated genes. (**A**) Percent integration in SPAD-associated genes in HEK293T cells is shown for two independent infections, with replicate 1 data plotted to the left. (**B**) Percent integration in SPAD-non-associated genes for the infections shown in panel (**A**). (**C**) Same as in panel (**A**) except that infections were conducted with Jurkat T cells. (**D**) Percent integration in SPAD-non-associated genes for the infections shown in panel (**C**). Asterisks show differences versus matched RIC (red) and DMSO (black) controls. *, *p* < 0.05; ***, *p* < 0.0001—see Appendix A for detailed statistical analyses.

**Figure 5 viruses-14-01883-f005:**
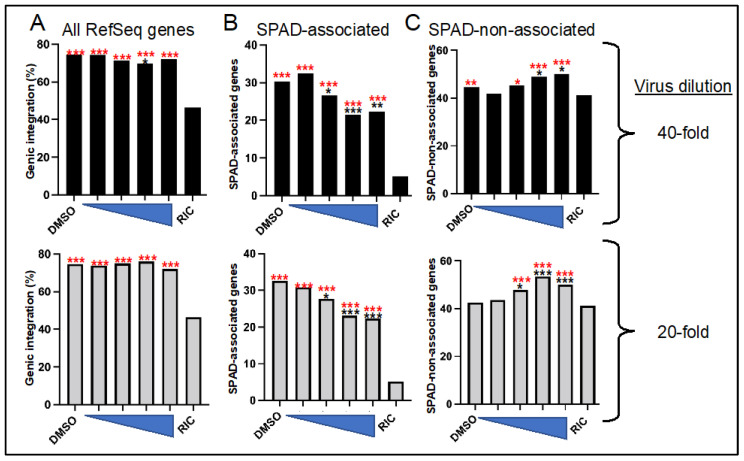
HIV-1 produced in the presence of CX014442 is defective for integration into SPAD-associated genes. (**A**) Percent integration into RefSeq genes as a function of CX014442 preexposure concentration (left to right, 31.25 nM to 250 nM, as indicated by triangle thickness). The upper panel is from virus diluted 40-fold prior to infection while the lower panel is from 1:20 diluted virus. (**B**) Same is in panel A except that percent integration into SPAD-associated genes was mapped. (**C**) Same as in panel A except that percent integration into SPAD-non-associated genes is shown. Asterisks show statistical differences versus matched RIC (red) or DMSO (black) controls. *, *p* < 0.05; **, *p* < 0.001; ***, *p* < 0.0001—see Appendix A for detailed statistical analyses.

**Figure 6 viruses-14-01883-f006:**
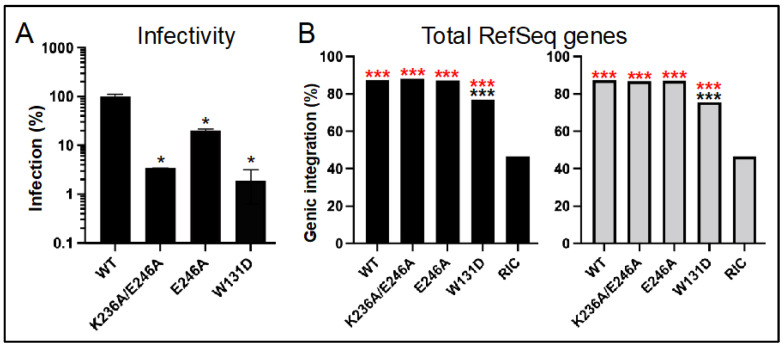
Class II IN mutant viral infectivities and genic integration targeting frequencies. (**A**) IN mutant HIV-Luc infection levels (average ± standard deviation from two independent experiments; within each experiment, luciferase values were determined for technical triplicate samples) were percent normalized to WT HIV-Luc, which was set to 100%. *, *p* < 0.01 (also see Appendix A). (**B**) Percent integration in RefSeq genes in HEK293T cells for the infections shown in panel A; replicate 1 data is plotted to the left. Asterisks show differences versus matched RIC (red) and WT HIV-Luc (black) controls. ***, *p* < 0.0001—see Appendix A for detailed statistical analyses.

**Figure 7 viruses-14-01883-f007:**
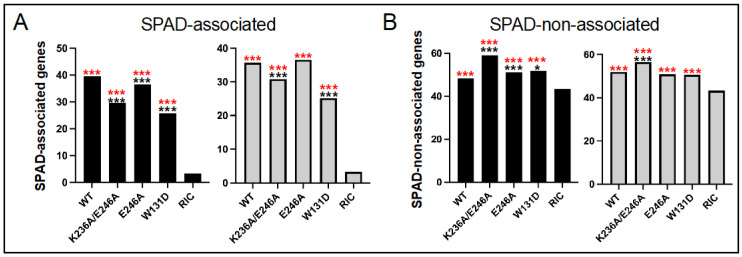
SPAD-associated gene integration defect of class II IN mutant viruses. (**A**) Frequencies of SPAD-associated gene integration for the noted viruses. The left graph is data from replicate 1. (**B**) Same as in panel (**A**), except for SPAD-non-associated genes. Asterisks show differences versus matched RIC (red) and WT HIV-Luc (black) controls. *, *p* < 0.01; ***, *p* < 0.0001—see Appendix A for detailed statistical analyses.

**Table 1 viruses-14-01883-t001:** Integration sites in BI-D-treated HEK239T cells ^1^.

Cell Type	BI-D	Replicate 1		Replicate 2	
		Number	In genes (%)	Number	In genes (%)
WT	– ^2^	1062	82.0	5100	80.4
	EC_70_	1971	69.1	4465	66.4
	EC_95_	2131	69.1	1969	70.4
HKO	–	16,266	81.7	10,767	82.4
	EC_70_	7299	67.3	6998	67.8
	EC_95_	4869	69.9	3154	68.9
LKO	–	3673	65.1	1755	63.9
	EC_70_	694	65.1	355	63.9
	EC_95_	246	55.7	303	55.4
DKO	–	1936	54.2	1639	55.5
	EC_70_	559	56.5	2170	54.7
	EC_95_	485	51.3	1171	53.5
RIC ^3^	n.a. ^4^	112,183	46.5	112,183	46.5

^1^ Shown are unique sites mapped and %integration in RefSeq genes. ^2^ –, infections conducted in presence of DMSO. ^3^ RIC, random integration control for MseI/BglII digestion. ^4^ n.a., not applicable.

**Table 2 viruses-14-01883-t002:** Integration sites derived from BI-D-treated virions ^1^.

Infected Cell	Producer Cell Treatment	Replicate 1				Replicate 2			
		Number	Refseq Genes	SPADg	SPAD-Nong	Number	Refseq Genes	SPADg	SPAD-Nong
HEK293T	DMSO	14,718	83.3	32.8	51.0	23,448	83.3	36.2	47.6
	BI-D	676	81.4	28.4	53.3	914	81.1	28.5	53.2
Jurkat	DMSO	14,583	86.3	34.9	51.8	27,044	85.9	35.2	51.2
	BI-D	812	85.2	29.9	55.7	654	83.3	26.9	56.9
RIC ^2^	n.a. ^3^	112,183	46.5	3.3	43.3	112,183	46.5	3.3	43.3

^1^ Shown are number of unique sites and %integration into RefSeq genes, SPAD-associated genes (SPADg), and SPAD-non-associated genes (SPAD-nong). ^2^ RIC, random integration control for MseI/BglII digestion. ^3^ n.a., not applicable.

**Table 3 viruses-14-01883-t003:** Integration sites derived from CX014442-treated virions ^1^.

CX014442 (nM)	1-to-40 Dilution				1-to-20 Dilution			
	Number	In Genes (%)	In SPADg (%)	In SPAD-Nong (%)	Number	In genes (%)	In SPADg (%)	In SPAD-Nong (%)
– ^2^	2725	74.4	30.3	44.5	3530	74.7	32.6	42.6
31.25	1282	74.3	32.5	41.8	1331	73.9	30.8	43.7
62.5	979	71.5	26.6	45.3	1028	75.0	27.7	47.9
125	1267	69.9	21.5	48.9	576	76.0	23.1	53.5
250	586	72.2	22.4	50.2	935	72.1	22.4	50.1
RIC ^3^	9,133,735	46.4	4.9	41.7	9,133,735	46.4	4.9	41.7

^1^ Shown are number of unique sites and %integration into RefSeq genes, SPAD-associated genes (SPADg), and SPAD-non-associated genes (SPAD-nong). ^2^ –, virus produced in presence of DMSO. ^3^ RIC, random integration control for shearing by sonication.

**Table 4 viruses-14-01883-t004:** Integration sites of class II IN mutant viruses ^1^.

IN	Replicate 1				Replicate 2			
	Number	In Genes (%)	In SPADg (%)	In SPAD-Nong (%)	Number	In Genes (%)	In SPADg (%)	In SPAD-Nong (%)
WT	7342	87.3	39.5	48.3	17,554	87.3	35.7	52.0
K236A/E246A	4754	88.1	29.7	59.1	2216	86.9	30.9	56.5
E246A	19,827	87.1	36.4	51.1	10,899	87.0	36.6	50.8
W131D	2347	77.0	25.8	51.7	3552	75.5	25.2	50.7
RIC ^2^	112,183	46.5	3.3	43.3	112,183	46.5	3.3	43.3

^1^ Shown are number of unique sites and %integration into RefSeq genes, SPAD-associated genes (SPADg), and SPAD-non-associated genes (SPAD-nong). ^2^ RIC, random integration control for MseI/BglII digestion.

## Data Availability

Raw Illumina sequences that were used to derive reported sites of HIV-1 integration (Table 1, Table 2 and Table 4) are available at the National Center for Biotechnology Information SRA (accession code PRJNA860913).

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
