# Peer review of "Allosteric Integrase Inhibitor Influences on HIV-1 Integration and Roles of LEDGF/p75 and HDGFL2 Host Factors"

_viruses, 2022, doi:10.3390/v14091883_

Round 1

Reviewer 1 Report

In this report, Singh et al report on the impact of the preclinical Allosteric Inhibitors of Integrase (ALLINIs) on HIV-1 integration into host genes and the roles of the host factors LEDGF/p75 and HDGF2 in these ALLINI-related integration retargeting effects. Using a system of LEDGF, HDGF, and double-knockout cell lines and the ALLINI BI-D, they demonstrate that HIV-1 produced in the presence of ALLINIs will integrate into host genes at normal frequencies, but the localization to nuclear speckles (SPAD) – the highly preferred integration target in the nucleus – are significantly impacted. They additionally discover that class II IN mutants have a similar property.

Overall, this is a well-written report contributing a very useful series of experiments to the literature and a logical extension of recent studies that together provide some understanding for the biology that underlies the ALLINI-related integration retargeting phenotype. The experiments are thoughtful with regards to existing literature and present well-controlled experiments with good statistical rigor. They reanalyze previously published data (CX014442 in SupT1 cells) to reaffirm the correlation of SPAD-associated genes with integration events.  The comparison of these observed effects to the class II mutants also provides a very useful insight.

The only suggested corrections are minor: this reviewer struggled a bit to follow the data figures throughout because of the lack of annotations on the panels about what experiment each panel represented. Addressing this will help the readers better appreciate the results.

Author Response

Please see pdf attachment

Reviewer 2 Report

The manuscript by Singh et al. entitled “Allosteric Integrase Inhibitor Influences on HIV-1 Integration and Roles of LEDGF/p75 and HDGFL2 Host Factors” investigates the integration retargeting of HIV-1 produced in the presence of ALLINIs, as well as the potential roles played by host factors LEDGF/p75 and HDGFL2. In this work, the authors demonstrate that HIV-1 produced in the presence of ALLINIs shows reduced integration into SPAD-associated genes. The authors also show that knocking out both LEDGF/p75 and HDGFL2 ablates the difference in HIV-1 genic integration in the presence vs. absence of ALLINI, suggesting that inhibition of these two host factors accounts for the ALLINI-induced integration retargeting. Additionally, the authors show that a number of Class II HIV integrase mutant viruses integrate less into SPAD-associated genes. The data presented in the manuscript are generally convincing, while the statistical analyses and presentation of the results can be improved. Throughout the manuscript, the authors plot two biological replicates side-by-side and annotate p-values calculated using Fisher’s Exact test (e.g. Fig 2B, 3B-C and etc.). Fisher’s Exact test is most commonly used for categorical data and not very suitable for the data generated from this study. Not to mention that it does not account for the experimental variance between the two biological replicates when used to compute p-values. Other methods such as t-test should be used for this purpose instead (even with n=2). And plotting two replicates into the one figure can greatly improve the clarity of the manuscript.

The integration results of random-integration control (RIC) seem to have been duplicated repeatedly in replicate 1 and 2 among Table 1, 2 and 4. Even though the RIC is generated in silico, duplicating it as “replicates” should be avoided unless explained in the text or legends.

Do LEDGF/p75 and HDGFL2 KOs demonstrate altered SPAD-associated gene integration compared with WT?

Author Response

Please see pdf attachment
